# Medium-to-Long-Term Immunogenicity of BNT162b2 mRNA COVID-19 Vaccine: A Retrospective Cohort Study

**DOI:** 10.3390/vaccines10030417

**Published:** 2022-03-10

**Authors:** Francesco Paolo Bianchi, Pasquale Stefanizzi, Cinzia Annatea Germinario, Giovanni Migliore, Luigi Vimercati, Andrea Martinelli, Annamaria Lobifaro, Giusy Diella, Angela Maria Vittoria Larocca, Silvio Tafuri

**Affiliations:** 1Department of Interdisciplinary Medicine, Aldo Moro University of Bari, 70124 Bari, Italy; dr.francesco.bianchi@gmail.com (F.P.B.); pasquale.stefanizzi@uniba.it (P.S.); cinziaannatea.germinario@uniba.it (C.A.G.); luigi.vimercati@uniba.it (L.V.); andrea.martinelli@uniba.it (A.M.); annamaria.lobifaro@policlinico.ba.it (A.L.); giusy.diella@uniba.it (G.D.); 2Hospital Direction, Bari Policlinico General Hospital, 70124 Bari, Italy; direzione.generale@policlinico.ba.it; 3Hygiene Department, Bari Policlinico General Hospital, 70124 Bari, Italy; laroccaangela1@gmail.com

**Keywords:** COVID-19, vaccine effectiveness, antibodies, documented infection, healthcare workers

## Abstract

To deal with the COVID-19 pandemic, a mass vaccination campaign was started in European countries on 27 December 2020. The first vaccine available to immunize healthcare workers (HCWs) was the BNT162b2 mRNA COVID-19 vaccine. While many studies have shown a high antibody response after the second vaccine dose, antibody persistence over the medium-to-long term has yet to be evaluated. The medium-to-long-term persistence of anti-SARS-CoV-2 antibodies was determined in a sample of fully vaccinated HCWs at Bari Policlinico General Hospital, Italy. This is a observational cohort study. HCWs who completed the immunization basal cycle were screened for anti-SARS-CoV-2 IgG on days 15, 30, 60, 90, and 120 after the second vaccine dose. At each time point, >99% of the screened HCWs were seroprotected. While the geometric mean titer initially declined over time, by 60 days the titer had stabilized. Older subjects seem to lose IgG faster than younger ones. The immunogenicity conferred by the vaccine provides further evidence that it is an essential weapon in efforts to bring the COVID-19 pandemic under control. Accordingly, strict measures should be implemented, ranging from the mandatory vaccination of HCWs to strong incentives aimed at achieving vaccination of the large majority of the overall population.

## 1. Introduction

COVID-19, the infectious disease caused by the novel coronavirus SARS-CoV-2, was declared a pandemic in early 2020, having reached global proportions [1]. By 28 July 2021, according to the World Health Organization (WHO), the amount of confirmed cases had reached ~194,000,000, including more than 4,170,000 deaths [2] while the European Centre for Disease Prevention and Control (ECDC) reported >34,000,000 cases and >750,000 deaths in Europe [3]. In Italy, there have been over 4,300,000 cases, including 139,000 cases in healthcare workers (HCWs), and 127,000 COVID-19-related deaths (fatality rate: 3%) [4].

To deal with the COVID-19 pandemic, a mass vaccination campaign was started in Europe on 27 December 2020. In Italy, healthcare workers (HCWs) were considered a high-risk group for which vaccination was prioritized, as recommended by the Center for Disease Control and Prevention (CDC) [5]. Indeed, HCWs are at a high risk of exposure to the virus due to their job activities and thus to the development of infection; furthermore, immunizing HCWs preserves healthcare capacity. The vaccine available to immunize health personnel was the BNT162b2 mRNA COVID-19 vaccine (Comirnaty; BNT162b2, BioNTech/Pfizer, Mainz, Germany/New York, NY, USA), the first anti-SARS-CoV-2 vaccine to be permitted by the European Medicines Agency (EMA). It is specified for individuals 12+ years of age and is administered in two doses dispensed at least 21 days apart [6,7].

In the phase III studies, the vaccine had a 95% efficacy in preventing infection, including severe complications [8]. Aside from transient local and systemic adverse reactions, no safety distresses were recognized [8]. Moreover, in 2021, a large phase IV study [9] investigated the effectiveness of the BNT162b2 mRNA vaccine in >1,000,000 Israeli inhabitants (50% fully vaccinated and 50% unvaccinated). The results showed that the vaccine prevented symptomatic illness with an effectiveness—measured as documented infection—of 94% 7 days after the second dose. The estimated vaccine effectiveness for documented infection after the first dose was 46% (95% CI = 40–51%) at 14–20 days and 60% (95% CI: 53–66%) at 21–27 days. In the follow-up period starting 7 days after the second dose, the effectiveness decreased to 92% (95% CI = 88–95%).

The immunogenicity of the vaccine was further demonstrated in other studies. In a 2020 phase I trial conducted in the United States [10], healthy adults 18–55 years of age and 65–85 years of age received either a placebo or the BNT162b2 mRNA vaccine. The vaccine elicited lower titers of antigen-binding IgG and virus-neutralizing responses in the older subjects than in the younger ones, but in both a higher vaccine dose elicited stronger antibody responses. A study published in 2021 [11] showed that the BNT162b2 mRNA vaccine induces anti-spike neutralizing antibodies associated with protective immunity, with the second dose of the vaccine inducing cross-neutralization of at least some of the circulating SARS-CoV-2 variants. A 2021 preprint of a brief report [12] described the antibody responses in 2015 HCWs who had received two doses of Comirnaty; 99.9% developed either seroconversion or a substantial increase in antibody titer. Nonetheless, the long-term immunogenicity conferred by the BNT162b2 mRNA vaccine is still unknown.

The purpose of our paper is to evaluate the medium-to-long-term persistence of anti-SARS-CoV-2 antibodies in a sample of healthcare providers at Bari Policlinico General Hospital who had been fully vaccinated with the Comirnaty vaccine. Our research was carried out in Apulia (Italy, almost 4,000,000 inhabitants), where, from February 2020 to 28 July 2021, 265,000 confirmed cases of COVID-19 and almost 7000 related deaths were recorded [13].

## 2. Materials and Methods

### 2.1. Study Design and Setting

This is an observational cohort study; it was conducted at Bari Policlinico General University Hospital (almost 1000 beds, almost 6000 healthcare providers), where ~180-hospital beds had been reserved for COVID patients and the emergency room was charged with the triage and care of those patients. The vaccination campaign for health personnel started on 27 December 2020; the scheduling and follow-up activities were coordinated by the Hygiene and Occupational Medicine departments.

HCWs requested vaccination by fulfilling an intranet online form. The Hygiene Department contacted the HCWs by mail or phone to schedule an appointment for vaccination. An appointment for the second dose was also made.

Vaccines were administered by public health physicians specialized in vaccinology. Two doses of the BNT162b2 mRNA vaccine were delivered intramuscularly in the deltoid muscle at least 21 days apart. Until April 2021, immunization prophylaxis in Italian HCWs was not mandatory. Informed consent was obtained at the time of vaccination. All vaccinated HCWs were followed up for 1 month to assess the development of any adverse effects.

Policlinico Bari General Hospital also adopted a specific procedure for the control and prevention of SARS-CoV-2 infection among its HCWs. All asymptomatic HCWs were screened every 14 days for SARS-CoV-2 infection using molecular tests of naso-pharyngeal swabs obtained as recommended by the WHO [14]. Fast-track access to molecular testing was ensured for HCWs with signs and symptoms of COVID-19 (fever, cough, ageusia, etc.). Infected HCWs were included in an active surveillance program conducted by Public Health physicians and followed up until they tested negative for the virus.

### 2.2. Participants

The population in our study included healthcare providers who had completed the basal immunization routine between 27 December 2020 and 31 March 2021. A list of eligible study participants was obtained from the Vaccination Clinic database. Subjects with a documented history of SARS-CoV-2 infection before or after enrollment were excluded from participation in the study.

Enrolled HCWs were screened on a voluntary basis and during routine occupational health follow-up exams to assess the presence of specific IgG anti-spike-protein antibodies using the Abbott IgG quantitative test (ARCHITECTURE SARS-CoV-2 IgG II Quant [15,16]). This automated, two-step chemiluminescent microparticle immunoassay allowed qualitative and quantitative determinations of IgG antibodies to SARS-CoV-2 in human serum and plasma. The sample, the antigen-coated paramagnetic microparticles SARS-CoV-2, and the assay diluent were dispensed together and incubated. The anti-SARS-CoV-2 IgG antibodies present in the sample bound to microparticles coated with SARS-CoV-2 antigens. Then, the mixture was washed. Antihuman IgG acridinium-labeled conjugate was added to create a reaction mixture and incubated. Following a wash cycle, Pre-Trigger and Trigger Solutions were added. The resulting chemiluminescent reaction was measured as a relative light unit (RLU). There was a direct relationship between the amount of IgG antibodies to SARS-CoV-2 in the sample and the RLU detected by the system optics. To ensure accurate results, serum and plasma samples must not contain fibrin, red blood cells, or other substance particulates. Frozen samples must be completely thawed before mixing them. For the dosage, it was recommended to perform a single analysis of each level of controls once every 24 h, each dosing day. The SARS-CoV-2 IgG II Quant assay uses a method of processing of the logistic fitting data of the 4-parameter curve (4PLC, Y-weighted) to generate a calibration curve and the results. The cutoff was 50.0 AU/mL. The analytical measurement interval was stated as 21 to 40,000 AU/mL.

HCWs included in this study were screened 15 (T0), 30 (T1), 60 (T3), 90 (T3), and 120 (T4) days after receiving the second dose. However, because follow-up was not mandatory, many of the fully vaccinated HCWs did not participate in the subsequent evaluations.

### 2.3. Statistical Analysis

The ultimate dataset was generated as an Excel worksheet that included data on sex, age, job type, and hospital unit, as well as serological data at each detection time (T0–T4). An anonymized analysis was executed using STATA MP17 software. Continuous variables were reported as the mean ± standard deviation and range, and categorical variables as proportions. The results of the serological analysis were expressed as the geometric mean titer (GMT) and its 95% confidence interval (95% CI). Repeated-measures mixed models were used to compare the serological data of the different groups, defined by: sex, age, job type (direct vs. indirect care of patients), ward (COVID vs. not COVID), and the detection time. Tests of simple effects were then performed to identify significant relationships between groups, time, and the interaction between group and time. For all tests, a two-sided *p*-value < 0.05 was considered to indicate statistical significance.

## 3. Results

The study population consisted of 3170 healthcare providers vaccinated with two doses of Comirnaty. The characteristics of the health personnel at enrollment are reported in Table 1. The median interval between the first and second vaccine doses was 22 days (interquartile range [IQR] = 22–23).

At T0, 99.6% (95% confidence interval [CI] = 97.7–99.9%; n = 237/238) of the tested HCWs were antibody-positive; at T1, all of the HCWs were positive (100.0%; 95% CI = 99.4–100.0%; n = 614/614); at T2, 99.8% (95% CI = 99.5–99.9%; 1763/1766) tested positive; at T3, all HCWs tested positive (100.0%; 95% CI = 99.6–100.0%; n = 831/831); and at T4, 99.7% (95% CI = 98.1–99.9%; 297/298) tested positive.

The trend in the GMT values per detection time is described in Figure 1. An initial decline in the titer was followed by what seemed to be a gradual leveling-off (*p* < 0.0001).

Table 2 showed the GMT when stratified for sex, age, job type (direct vs. indirect care of patients), ward (COVID vs. not COVID), and detection time (Appendix A).

With respect to sex, significant differences between sexes (*p* = 0.028) and in the detection time (*p* < 0.0001), but not in the interaction between sexes and time (*p* = 0.160), were determined. The results of the tests of simple effects to explain the significant interactions are shown in Appendix A.

The differences in the GMT titer between age classes and with respect to detection time were significant (both *p* < 0.0001). The interaction between age and time was also significant (*p* < 0.0001). The results of the tests of simple effects to explain the significant interactions are shown in Appendix A.

In the analysis of the GMT according to job type (direct or indirect care of patients), the differences between groups (*p* = 0.004) and with respect to the detection time (*p* < 0.0001), but not the interaction between job type and time (*p* = 0.320), were significant. The results of the tests of simple effects to explain the significant interaction are presented in Appendix A.

For HCWs working in non-COVID vs. COVID wards, the difference in the detection time was significant (*p* < 0.0001), but the differences between groups and for the interaction between group and time were not (*p* = 0.937 and *p* = 0.870, respectively). The results of the tests of simple effects to explain the significant interaction are presented in Appendix A.

During the follow-up, no serious and/or long-term adverse reactions (death, hospitalization, severe allergic reactions) were reported. The safety of the vaccine is the subject of a manuscript currently in preparation.

## 4. Discussion

Our study of a sample of healthcare providers showed that the BNT162b2 mRNA vaccine led to a very high humoral immunity (>99%) over a period of 120 days. Although the GMT initially declined, it was mostly stabilized by T3. The causes and consequences of this decline should be evaluated in further studies. Our results were to some extent consistent with those of a 2021 Italian study [17] of a sample of fully vaccinated HCWs who were evaluated for anti-COVID-19 IgG titers before the first vaccine dose, 21 days after the first dose, and 30 days after the second one. The authors found an increase in the antibody titer over time but they did not evaluate the changes at longer intervals after the second dose.

According to the repeated-measures mixed model, the immunological response and the duration of circulating antibodies were better in females than in males. Previous studies of sex-based differences in response to vaccines or infection [11,18] consistently showed a more effective immune response to immunization in females. However, the role of sex in the response to anti-SARS-CoV-2 vaccines remains to be determined.

Our finding of a better immune response in younger than in older HCWs; indeed, as showed by Table 2, trends were similar among the age classes, but older HCWs showed a lower antibody titer after the immunization basal cycle compared to younger ones. This was in agreement with two recent studies [18,19] which reported that after the first vaccine doses, IgG antibody levels were significantly lower in the older age group (55–65 years) compared to younger age groups (20–34 and 35–44 years); the main differences between the two groups were likely a consequence of immunosenescence, which described the reduced adaptive immune responses in the elderly. Müller L et al. [20] reported differences between the antibody responses raised after the first and second BNT162b2 vaccination, in particular lower frequencies of antibodies in the elderly subjects; the authors suggested that older persons need to be closely monitored and may require earlier revaccination and/or an increased vaccine dose to ensure stronger long-lasting immunity and protection.

HCWs whose work involved direct contact with patients had better immunity at all times, whereas there were no differences in HCWs working in non-COVID vs. COVID wards. This finding was not unexpected, given the high level of protection ensured for Bari Policlinico healthcare personnel to reduce their biological risk, including the mandatory use of PPE regardless of the assigned hospital unit.

The strengths of our study were its considerable sample size and lengthy follow-up period. The most important limitation was the low rate of timely follow-up, due to the nonmandatory nature of the evaluations and to the work commitments of the HCWs. This issue was addressed by our use of the repeated-measures mixed model [21]. Another limitation is that our study did not evaluate the cellular response, nor neutralizing antibodies. Additional studies are required to establish the immunological and cellular response to the BNT162b2 mRNA vaccine in different populations and over an even lengthier follow-up period.

Thus far, the data were consistent with the absolute effectiveness of the BNT162b2 mRNA vaccine in preventing SARS-CoV-2 infection and COVID-19 disease. As already noted in a brief report published in 2021 by our research team [22], the vaccine was highly effective against asymptomatic infection (>90%) in Bari Policlinico HCWs. The medium-to-long-term persistence of IgG, as evaluated in the present study, provided further evidence that the vaccine is an essential tool in bringing the pandemic under control, including the elimination of outbreaks in the nosocomial setting. While vaccination hesitancy among Italian HCWs has thus far been tolerated, on 31 March 2021 the Italian government made anti-SARS-CoV-2 vaccination semicompulsory, with a penalty of salary suspension—although the practical effects of this regulation are only now beginning to be realized. As reported elsewhere [23], vaccination refusal is a widespread and difficult problem to solve, both in the general population and specifically in HCWs.

A broader confirmation of our finding of a decline in the GMT could lead to a policy of periodic revaccination, remarkably for healthcare providers. Indeed, the UK’s Department of Health and Social Care recently declared the rollout of an anti-SARS-CoV-2 booster vaccine, planned for the beginning of fall, in order to safeguard the most exposed members of the population ahead of wintertime, when a further surge in cases is expected [24]. Moreover, the CEO of Pfizer, one of the companies that developed and produced the BNT162b2 mRNA vaccine, also commented that a booster dose ~12 months after the initial vaccination basal routine will “likely” be needed. Additional follow-up time and scientific research will be desirable to define whether this is indeed the case.

In conclusion, the demonstrated immune response, effectiveness, and safety of the vaccine provide further evidence of its importance in combating the COVID-19 pandemic. However, at least in Italy, too many healthcare providers and large sectors of the general population are still not vaccinated. Considering that the conclusion of the pandemic is not yet in sight, stricter measures, such as mandatory vaccination for HCWs, should be implemented together with stronger vaccination incentives for the overall population. Finally, particular attention must be paid to the older age groups, considering that the lower immune response and the decrease in antibodies over time increase the risk of COVID-19 infection in this category with a high risk of complications.

## Figures and Tables

**Figure 1 vaccines-10-00417-f001:**
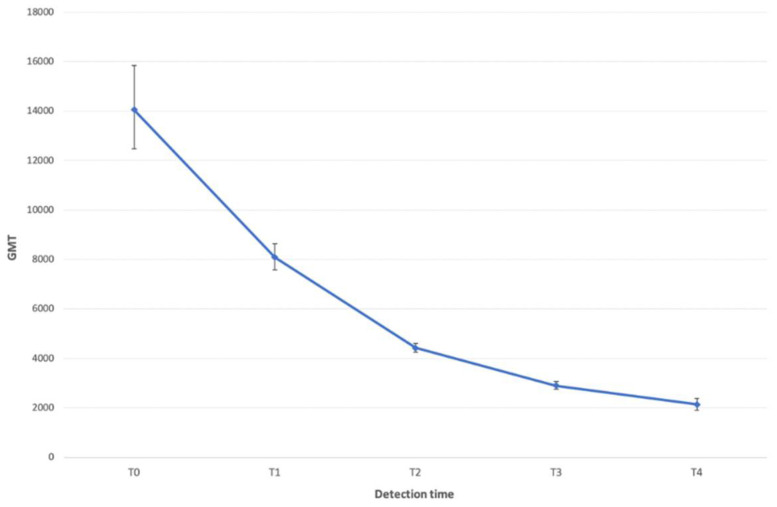
Trend in the GMT of anti-COVID-19 IgG in the screened HCWs. The one-way ANOVA test showed a statistically significant trend (*p* < 0.0001).

**Table 1 vaccines-10-00417-t001:** Characteristics of the study population at baseline.

Variable	Values
Age (years); mean ± SD (range)	44.7 ± 12.6 (22–69)
Female; n (%)	1948 (61.5)
Professional category; n (%) • Physician • Nurse • Auxiliary staff • Other HCWs	1176 (37.1) 1022 (32.2) 490 (15.5) 482 (15.2)
Medical area; n, (%) • Service • Surgery • Clinic	1030 (32.5) 1406 (44.4) 734 (23.1)
Ward; n, (%) • Not COVID • COVID	2435 (76.8) 735 (23.2)

**Table 2 vaccines-10-00417-t002:** Trend in the GMT (95% CI) of anti-COVID-19 IgG in screened HCWs, per group.

Variable	T0	T1	T2	T3	T4
Sex					
Female	14,524.8 (12,335.1–17,103.3)	8646.0 (7933.2–9422.9)	4733.6 (4520.9–4956.4)	3082.2 (2889.1–3288.3)	2252.8 (1973.0–2572.3)
Male	13,551.4 (11,346.6–16,185.4)	7393.9 (6692.3–8169.1)	3970.4 (3699.8–4260.8)	2584.1 (2347.4–2844.7)	1906.1 (1560.0–2328.9)
Age class					
18–35 years	17,802.7 (13,647.2–23,117.7)	10,302.4 (9307.7–11,403.5)	5854.2 (5527.8–6199.9)	3653.8 (3376.1–3954.5)	2722.0 (2417.0–3065.4)
36–50 years	14,459.8 (12,375.0–16,985.9)	8238.3 (7314.7–9278.5)	4278.1 (4023.8–4.548.5)	2748.8 (2504.8–3016.6)	1795.9 (1486.5–2169.7)
51–70 years	11,508.3 (9368.8–14,136.4)	6603.0 (5923.9–7359.9)	3703.3 (3445.7–3980.2)	2436.0 (2205.8–2690.2)	1836.9 (1434.7–2351.9)
Job type					
Direct care of patients	14,253.4 (12,328.2–16,479.3)	8737.8 (8067.8–9463.4)	4700.2 (4496.3–4913.2)	2992.5 (2818.5–3177.3)	2323.7 (2047.9–2636.6)
Indirect care of patients	13,558.9 (10,955.9–16,780.5)	7128.7 (6381.1–7964.0)	3856.2 (3561.6–4175.3)	2583.8 (2284.4–2922.5)	1725.2 (1384.3–2150.1)
Ward					
COVID	14,452.9 (12,661.1–16,498.3)	8106.4 (7528.2–8728.0)	4301.3 (4107.0–4504.9)	2904.1 (2731.2–3088.0)	2061.1 (1772.0–2397.3)
Not COVID	12,959.1 (9887.4–16,985.2)	8006.6 (6962.3–9207.5)	4823.6 (4477.3–5196.7)	2880.4 (2569.3–3229.2)	2269.8 (1958.1–2631.1)

## Data Availability

Data are available on request due to restrictions, e.g., privacy or ethical.

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
