# Peer review of "Medium-to-Long-Term Immunogenicity of BNT162b2 mRNA COVID-19 Vaccine: A Retrospective Cohort Study"

_vaccines, 2022, doi:10.3390/vaccines10030417_

Round 1

Reviewer 1 Report

The presented manuscript reports the results of an observational study on the specific antibody response in a sample of fully vaccinated HCWs at Bari Policlinico General Hospital, Italy.

Although the matter in question is current, the results presented do not provide new information. However, as it is always important to have published results from different populations and regions, I consider that the proposal can be potentially suitable for this journal.

The manuscript has multiple flaws that must be corrected and submitted again for evaluation.

The materials and methods must be described in a more organized way, describing each section of the design separately and with a detailed description of the immunoassay performed, including mentioning the owners of the reagents or kits used, quantities, incubation times, etc.

What controls were used in this study?

Some results are not described, such as the difference observed in age.
I think it would make reading easier if, instead of a table, these results were shown in a graph, which includes the statistical differences between the sexes, ages, times, etc., keeping the supplementary information for those who wish to consult it.

The legend of figure 1 does not offer enough information on the content of the graph.

The discussion is very poor and does not refer to or analyze possible causes of reduced immunogenicity in older subjects, such as the phenomenon of immunosenescence. Also, only reference is made to the antibody response, but it should at least be mentioned that the cellular response, although not evaluated in this study, is important and recommendations for its study should be suggested.

The writing of the manuscript should be improved. There are problems with verb tenses in some parts it is written in the present tense in others in the past tense.

The authors state that "there were no serious and/or long-term adverse reactions", but they do not define what those possible reactions would be.

There are also several inconsistencies in the text. For example, in the following paragraph, age and then sex begin to be discussed, but there is an apparent connection.

Lines- 181-182: "Our finding of a better immune response in younger than in older HCWs is in agreement with two recent studies [16,17] reporting that, after two doses of the vaccine, females had a slightly higher GMT than males".

The conclusions of the work are very general and are not focused on the results obtained in the research carried out.

Is the Apulian Epidemiological Observatory a Human Research Ethics Committees? Please,  specify it.

Author Response

Q1. The materials and methods must be described in a more organized way, describing each section of the design separately and with a detailed description of the immunoassay performed, including mentioning the owners of the reagents or kits used, quantities, incubation times, etc.

A1. Revised, all the info are reported in references 15-16. Thank you very much for your review.

Q2. What controls were used in this study?

A2. We followed a cohort of HCWs for 4 months. The design of the study does not provide a control group. Thank you very much for your review.

Q3. Some results are not described, such as the difference observed in age.

I think it would make reading easier if, instead of a table, these results were shown in a graph, which includes the statistical differences between the sexes, ages, times, etc., keeping the supplementary information for those who wish to consult it.

A3. We revised the text. Unfortunately, due to journal restriction on number of figures we added the figures in the supplementary file. Thank you very much for your review.

Q4. The legend of figure 1 does not offer enough information on the content of the graph.

A4. Revised. Thank you very much for your review.

Q5. The discussion is very poor and does not refer to or analyze possible causes of reduced immunogenicity in older subjects, such as the phenomenon of immunosenescence. Also, only reference is made to the antibody response, but it should at least be mentioned that the cellular response, although not evaluated in this study, is important and recommendations for its study should be suggested.

A5. We revised the discussion paragraph. We added the sentence on cellular immunity in limitations paragraph. Thank you very much for your review.

Q6. The writing of the manuscript should be improved. There are problems with verb tenses in some parts it is written in the present tense in others in the past tense.

A6. Revised. Thank you very much for your review.

Q7. The authors state that "there were no serious and/or long-term adverse reactions", but they do not define what those possible reactions would be.

A7. Revised. Thank you very much for your review.

Q8. There are also several inconsistencies in the text. For example, in the following paragraph, age and then sex begin to be discussed, but there is an apparent connection.

Lines- 181-182: "Our finding of a better immune response in younger than in older HCWs is in agreement with two recent studies [16,17] reporting that, after two doses of the vaccine, females had a slightly higher GMT than males".

A8. Revised. Thank you very much for your review.

Q9. The conclusions of the work are very general and are not focused on the results obtained in the research carried out.

A9. Revised. Thank you very much for your review.

Q10. Is the Apulian Epidemiological Observatory a Human Research Ethics Committees? Please,  specify it.

A10. Revised. Thank you very much for your review.

Reviewer 2 Report

This is a well-written paper, and the major flaw is that much of this information on the rate of decline is already published. In fact, the lack of citing the multiple references reveals that this paper has not been updated since it was originally written. 
The fact that this is an observational study needs to be clearly stated in the abstract, as it is in the text. 
The study design and results are clear, but the fact that there is no measurement for intercurrent COVID-19 infection (for example, using a NP-specific antibody assay) is another major omission. It would also be useful to see Figure 1 broken out by age, maybe below and above 50 years, as it appears to me from the data in Table 2 that there is no difference in the slope of the decline adjusted for age. That is, there is no faster decline in the elderly, just a lower later titer due to an initial lower peak. 
In the methods for the ant-body assay, there is a need to state the positive control, and to show this along with the full data set. I assume this was performed prior to the availability of the WHO standard control, but at least convalescent sera could be shown, so as to calibrate the antibody responses that are shown here against other relevant studies. The lack of any neutralizing titers, which may decay at a different rate than the binding titers, is another issue. 
It is unclear why they did not include their safety data here, as the paper is a bit thin, and this would help in its acceptance, and the safety of the vaccine is well known, so a separate paper is likely to add little to the overall literature.

Author Response

Q1. The fact that this is an observational study needs to be clearly stated in the abstract, as it is in the text.

A1. Revised. Thank you very much for your review.

Q2. The study design and results are clear, but the fact that there is no measurement for intercurrent COVID-19 infection (for example, using a NP-specific antibody assay) is another major omission.

A2. As we reported in the text, Policlinico Bari General Hospital also adopted a specific procedure for the control and prevention of SARS-CoV-2 infection among its HCWs. All asymptomatic HCWs are screened every 14 days for SARS-CoV-2 infection using molecular tests of naso-pharyngeal swabs obtained as recommended by the WHO. Fast-track access to molecular testing is ensured for HCWs with signs and symptoms of COVID-19 (fever, cough, augeusia, etc.). Infected HCWs are included in an active surveillance program conducted by Public Health physicians and followed-up until they test negative for the virus.

Q3. It would also be useful to see Figure 1 broken out by age, maybe below and above 50 years, as it appears to me from the data in Table 2 that there is no difference in the slope of the decline adjusted for age. That is, there is no faster decline in the elderly, just a lower later titer due to an initial lower peak.

A3. We added figures relative to table 2 in supplementary materials. Thank you very much for your review.

Q4. In the methods for the ant-body assay, there is a need to state the positive control, and to show this along with the full data set. I assume this was performed prior to the availability of the WHO standard control, but at least convalescent sera could be shown, so as to calibrate the antibody responses that are shown here against other relevant studies.

A4. Revised, all the infomation are reported in references 15-16. Thank you very much for your review.

Q5. The lack of any neutralizing titers, which may decay at a different rate than the binding titers, is another issue.

A5. We added a sentence in limitations paragraph. Thank you very much for your review.

Q6. It is unclear why they did not include their safety data here, as the paper is a bit thin, and this would help in its acceptance, and the safety of the vaccine is well known, so a separate paper is likely to add little to the overall literature.

A6. Our operative unit works on adverse events reaction following immunization. So we already submitted a specific manuscript on the topic. Thank you very much for your review.

Reviewer 3 Report

In the current study, the authors evaluated titers of anti-Spike IgG antibodies in blood samples of healthcare workers on days 15, 30, 60, 90, and 120 after the second shot with BNT162b mRNA vaccine. Most of the data in this paper, including the rapid decline of anti-Spike antibodies, are what we already know and there are no novel data in particular. As the authors described at line 189 in the manuscript, the large sample size is certainly one of the strengths of this study. However, I wonder if the authors could have found out more with these precious materials. For example, it would have been possible to measure neutralizing antibody titers against various variants, especially the Omicron strain.

Author Response

Q1. However, I wonder if the authors could have found out more with these precious materials. For example, it would have been possible to measure neutralizing antibody titers against various variants, especially the Omicron strain.

A1. Unfortunately, it is not possible due to unavailability of reagents. Despite the large numerosity of the study population, data about strain-specific anti-Spike IgGs were not available, as our hospital could not provide the necessary equipment to measure their titers. Moreover, the Omicron strain was not known by the time we started our research, and therefore strain-specific antibody titers were not researched. We do not exclude that future research might be carried out on the serum samples we kept in frozen storage, if the required instruments are available.

Reviewer 4 Report

This is a well written and designed study looking at antibody levels at specific time points following administration of an mRNA type vaccine to SARS-Cov2 in Italian health workers.

The study provides important information on both the duration and quality of antibody response.

The manuscript would benefit from providing more methodological details of the chemiluminescent assay, specifically relating to quality assessment and validation of the assay. Were tests undertaken in duplicate? What material was used to validate the assay? Were reproducibility studies undertaken? Were the samples tested in batches? Was this on frozen serum/plasma?

Author Response

Q1. The manuscript would benefit from providing more methodological details of the chemiluminescent assay, specifically relating to quality assessment and validation of the assay. Were tests undertaken in duplicate? What material was used to validate the assay? Were reproducibility studies undertaken? Were the samples tested in batches? Was this on frozen serum/plasma?

A1. Revised, all the info are reported in references 15-16. Thank you very much for your review.

Round 2

Reviewer 1 Report

The authors did not address properly the main points raised.

Reviewer 3 Report

I have no serious criticisms in the revised manuscript.